# South Asian origin and global transmission history of *Mycobacterium tuberculosis* lineage 4

Bharkbhoom Jaemsai,[1,2] Prasit Palittapongarnpim,[1,2] Pakorn Aiewsakun[1,2]

**ABSTRACT** The origin and transmission history of *Mycobacterium tuberculosis* lineage 4 (MTB L4) has been extensively studied. However, different studies yielded different results; thus, the evolutionary history of MTB L4 still remains a subject of debate. Recently, a substantial amount of whole-genome sequencing data of MTB has become available, providing new data from diverse geographical locations worldwide. This study analyzed the most comprehensive global collection of MTB L4 genomes to date ($n$ = 11,154), including new sequences from previously under-represented regions, to re-examine the bacterial evolutionary history. Our results suggest that the bacteria likely emerged and diversified in South Asia during the 12th and 13th centuries and then spread to various Old-World regions and the Americas between the 15th and 16th centuries, before undergoing extensive intercontinental transmissions starting from the 17th century onward. The effective population size of MTB L4 as a whole was estimated to expand steadily throughout its evolutionary history until the mid-20th century when a sharp drop occurred, coinciding with the introduction of antibiotics and significant improvements in human living conditions. Interestingly, this pattern was consistently observed across all major sub-lineages, indicating a broad impact of these factors on MTB L4 as a whole. Altogether, our results offer new refined insights into how anthropological changes might have shaped the bacterial evolutionary history and ultimately its global geographical distribution we observe today.

**IMPORTANCE** Contrary to previous studies, our analysis suggests that *Mycobacterium tuberculosis* lineage 4 (MTB L4) likely emerged and diversified in South Asia during the 12th and 13th centuries. It then spread to both the Americas and other Old-World regions between the 15th and 16th centuries, followed by extensive intercontinental transmissions beginning in the 17th century. These findings suggest that South Asia, one of the main crossroads of historical trade networks, might have played a pivotal role in promoting the early intercontinental spread of MTB L4. The effective population size of all major MTB L4 sub-lineages was inferred to increase steadily until the mid-20th century, after which a decline was observed. This decline coincides with the advent of antibiotics and improved living conditions, suggesting a wide impact of these factors on the entire bacterial population.

**KEYWORDS** *Mycobacterium tuberculosis* lineage 4, phylogeography, phylodynamics

M ycobacterium tuberculosis (MTB), the causative agent of tuberculosis (TB), is an ancient bacterial pathogen that has been circulating in human populations for thousands of years (1). MTB belongs to the *Mycobacterium tuberculosis* complex (MTBC), which, together with *M. Africanum* (MAF), constitutes the 10 currently known human-adapted lineages (MTB L1-L4 and L7-L8 and MAF L5-L6 and L9-L10) (2, 3), and they form separate phylogenetic clusters from animal-adapted strains within the complex, including *M. microti*, *M. pinnipedii*, *M. orygis*, *M. caprae*, and *M. bovis* (4). As obligate

Address correspondence to Pakorn Aiewsakun, pakorn.aie@mahidol.ac.th.

The authors declare no conflict of interest.

See the funding table on p. 14.

human parasites, the transmission histories, evolution, and global distributions of MTB have often been hypothesized to be largely driven and shaped by human migrations and interactions (5).

Among all human-adapted lineages within the MTBC, MTB L4 is the most geographically widespread cause of TB in humans (6), and its evolutionary history has been extensively studied; however, different studies have proposed different origins for the bacteria. By assuming that L5 and L6 co-diversified in parallel with its human host ~70,000 years ago, Comas et al. (7) estimated the age of the most recent common ancestor (MRCA) of the entire MTBC to be ~72,000 years old and, indirectly, estimated L4 to be ~30,000 years old. With a more direct tip-dating analysis, Bos et al. (8) used three Peruvian historical animal-adapted *M. pinnipedii* genomes (but which were obtained from human skeletons, dated between 1028 and 1280 CE) to provide temporal structure to their data and estimated the MRCA of the MTBC to be just about 4,064 (2,951–5,339) years old, along with an age estimate of MTB L4 of ~1,254–2,343 years old (fourth century BCE to eighth century CE), challenging the results reported by Comas et al. (7). Using a similar approach but with four historical human-adapted MTB L4 genomes obtained from Hungarian mummified remains (dated between 1787 and 1805 CE), Kay et al. (9) directly dated MTB L4's MRCA to the fourth century (396 CE, 40–662 CE), aligning more with Bos et al.'s results (8). By combining the Peruvian and Hungarian historical samples together in one tip-dating analysis with one additional historical Swedish MTB L4 genome (1679 CE), Sabin et al. (10) estimated the time to MRCA (tMRCA) of the MTBC to be ~3,258 (2,190–4,501) years, with that of L4 being ~1,118–2,468 years (fifth century BCE to ninth century CE). They also performed an L4-specific analysis and obtained similar results, dating the bacteria's MRCA to the 6th century (1,445 years before present) with a relatively large uncertainty ranging from the 1st century BCE to the 11th century CE (929–2,084 years before present). By using the rate reported by Kay et al. (9) to time calibrate their phylogeny, O'Neill et al. (11) suggested that MTB L4 likely originated around the fourth century BCE to fourth century CE (77 CE, 368 BCE–362 CE), likely in the region of the Mediterranean basin (Africa or Europe) with their phylogeographic analyses. With a direct tip-dating analysis using three high-quality historical MTB L4 genomes (1787–1797 CE) reported by Kay et al. (9) and several other relatively old genomes from Denmark (1961–1999 CE), Brynildsrud et al. (12) dated the origin of L4 to be in the 10th–13th centuries CE (1096 CE, 955–1231 CE), which is the youngest age estimate among all, and proposed specifically a European origin of MTB L4 with their phylogeographic analyses. Given all these different estimated timescales and geographical origins, the origin and transmission history of MTB L4 still remains a subject of debate.

Recent global efforts have sequenced a large number of MTB isolates from many previously under-sampled countries, including India (13), Indonesia (14), Thailand (15), and several countries in Africa and South America (16). These new data offer an opportunity to improve our understanding of the global transmission history, genetic diversity, and population dynamics of MTB L4. In this study, we analyzed, to our knowledge, the largest and most comprehensive global collection of whole-genome sequencing (WGS) data of MTB L4 ($n$ = 11,154) to estimate their evolutionary history, incorporating new data from many previously under-represented countries. Our results provide new insights into its relationship with their human hosts.

## RESULTS AND DISCUSSION

### Global collection of MTB L4 WGS data analyzed in this study

This study analyzed whole-genome short-read sequences of 11,154 MTB L4 samples reported from 106 countries (Fig. 1A; Table S1), which we tentatively grouped into nine major geographical regions, including Australia, East Asia, Southeast Asia, South Asia, Central-West Asia, Europe Africa, North America, and South America (Fig. 1B; Fig. S1). Among these, three genomes were of ancient MTB L4 samples. One was extracted from a calcified lung nodule obtained from a naturally mummified remain buried at Lund Cathedral, Sweden (LUND1: SRR11524778), dating back to the 17th century (10).

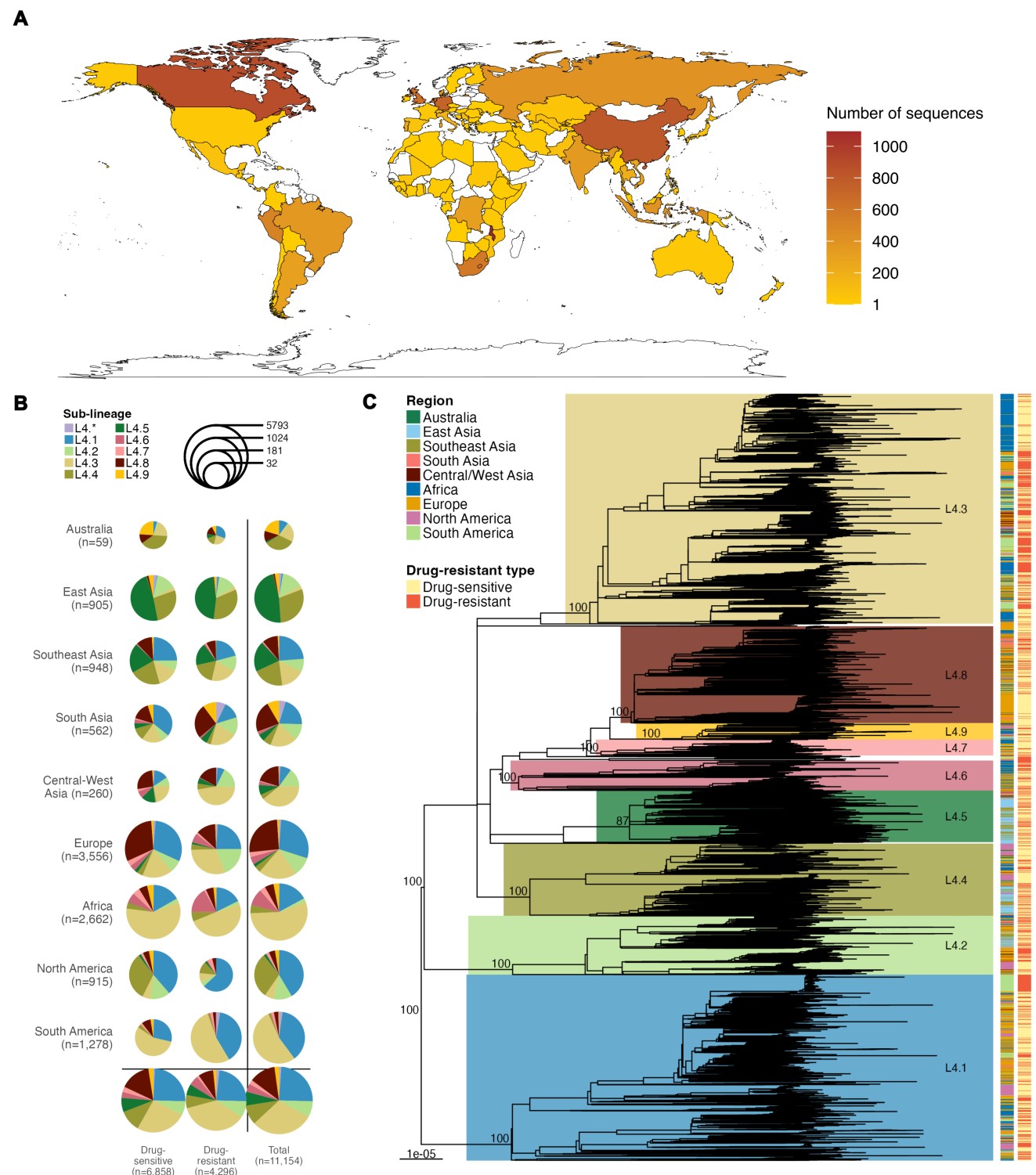

**FIG 1** Overview of MTB L4 WGS data (*n* = 11,154) analyzed in this study. (A) Geographical origins of the 11,154 MTB L4 whole-genome sequences analyzed in this study. (B) Distribution of samples by geographical source and genotypic drug-resistant type. Sizes of the pie charts represent the number of samples, and colors indicate sub-lineages (see key). The term "L4.*" collectively refers to samples that could not be assigned to the nine major sub-lineages defined by Coll et al. (L4.1–4 and L4.6–9) (17), Mokrousov et al. (L4.5.1) (18), and Ajawatanawong et al. (L4.5.2–3) (15). (C) Maximum likelihood tree of MTB L4 estimated from a whole-genome SNP alignment (356,934 sites). The tree was estimated using the GTR + I + Γ(4) nucleotide substitution model and rooted using an MTB L1 sample (accession number: SRR8375623, not shown in the figure). Clades of the nine major sub-lineages are highlighted, with clade bootstrap support values. The clade support values were computed based on 1,000 bootstrap trees using the ultrafast bootstrap approximation method implemented in IQ-TREE2 (19). Sampling location and genotypic drug-resistant type are indicated in the two columns on the right side of the tree, from left to right.

The other two were extracted from 18th century mummies housed in a crypt in the Dominican church of Vác in Hungary (B80: ERR651003, and B92: ERR651004) (9). Their sampling dates were assumed to be the dates of their human host death (LUND1: 1679 CE, B80: 1805 CE, B92: 1787 CE). The rest were from contemporary samples collected between 1961 and 2019 CE. While there are several more historical MTB L4 genomes available (9), these three genomes were of the highest quality ones that had previously been determined to be suitable for phylogenetic analysis (10). The majority of samples in our data set (11,026/11,154 = 98.85%) could be assigned to the nine major sub-lineages of MTB L4 based on established classification schemes (15, 17, 18) (Table S1 and Fig. S1). The ancient sample B80 was assigned to L4.1, and both LUND1 and B92 were assigned to L4.8, consistent with previous reports (9, 10).

Of the total sample analyzed, 61.48% (6,858/11,154) were genotyped as drug sensitive (DS), while 38.52% (4,296/11,154) were genotyped as drug resistant (DR) (see Table S1 for detailed DR-genotype breakdown). Notably, 27.86% of our data set was multidrug-resistant (MDR)/rifampicin-resistant (RR)/pre-extensively drug-resistant (pre-XDR) TB samples (Table S1), while the global rate of MDR/RR/pre-XDR TB is just about 3.70% (computed based on the global total estimate of 10.8 million TB cases and 400,000 cases of MDR/RR/pre-XDR TB reported for the year 2023 [20]). This large discrepancy was likely due to the biased sequencing efforts of DR-TB isolates to improve DR-TB surveillance and guide personalized treatment strategies.

## Overall phylogenetic structure of MTB L4

Maximum likelihood (ML) phylogenetic analysis was conducted to examine the overall phylogenetic structure of MTB L4. However, as noted above, our data set was enriched with DR samples. Genes associated with DR-conferring mutations are known to show relatively high genetic diversities compared to other genes (21–25), and convergent evolution of DR-conferring mutations is also known to be common in MTB (26–28). This, therefore, can potentially bias evolutionary rate estimation and falsely group DR samples together into clades on the tree due to their convergent evolutionary changes. To minimize these potential impacts of DR-MTB sampling bias on our phylogenetic inference, this analysis (and other subsequent phylogenetic analyses in this study) was thus conducted without including known DR-conferring mutation sites and their associated genes in the sequence alignment (see Materials and Methods for more details).

Overall, our phylogenetic analysis (Fig. 1C) showed that all nine major sub-lineages formed their own well-supported clades (bootstrap supports >90 for all clades), and the inferred phylogenetic relationships were consistent with previous findings (3, 12, 26). Samples that could not be assigned to any major sub-lineage (128/11,154 = 1.15%) formed multiple small phylogenetic clusters basal to the clades of the nine major sub-lineages. DR samples were found to distribute across geographical regions and the phylogeny (Fig. 1C), indicative of a widespread and independent occurrence of drug resistance in the bacterial evolutionary history and various geographical locations, consistent with previous results (26–28). This likely resulted from a combination of independent local emergences and cross-regional transmissions of DR strains.

## Evolutionary rate and timescale estimation

We time-calibrated the global MTB L4 phylogeny in two steps. We first performed Bayesian tip-dating analysis using BEAST v1.10.4 to best estimate the evolutionary rate and timescale of the bacteria using only samples with relatively precise collection dates (at least at the level of collection year). We then used the obtained rate to inform ML tip-dating analysis of the global tree, which then also included samples with collection date ranges and those without collection dates, while properly accounting for their collection date uncertainty.

The initial Bayesian tip-dating analysis was computationally very expensive, and to make the computation feasible, the analysis was performed on a down-sampled data

set of 377 sequences while still preserving all major sub-lineages and all geographical regions (see Supplemental methods). The three ancient samples were retained within the data set to preserve as much temporal signal as possible. This, indirectly, also reduced potential biases in the rate estimation potentially introduced by over-representation of samples from certain outbreaks with many isolates reported.

Root-to-tip regression analysis confirmed the presence of a sufficiently strong temporal signal in the data set (regression slope = $1.02 \times 10^{-7}$ substitutions per genome site per year [s/n/y], $P$ value < 0.001; Fig. 2A). The data set was also tested for the time-dependent rate phenomenon (TDRP), that is, in an evolutionary analysis of a sufficiently large timescale, longer-term average evolutionary rate estimates will tend to be lower than those computed over shorter timescales mainly due to purifying selection (29, 30), but no significant evidence for the TDRP was found (Fig. S2; slope's $P$ value = 0.9673), justifying the standard tip-dating analysis.

Various combinations of tree and clock prior models were explored (see Supplemental methods), yielding comparable rate estimates across different model combinations (Fig. 2B), but tMRCA estimates were slightly different (Fig. 2C). Model selection analysis

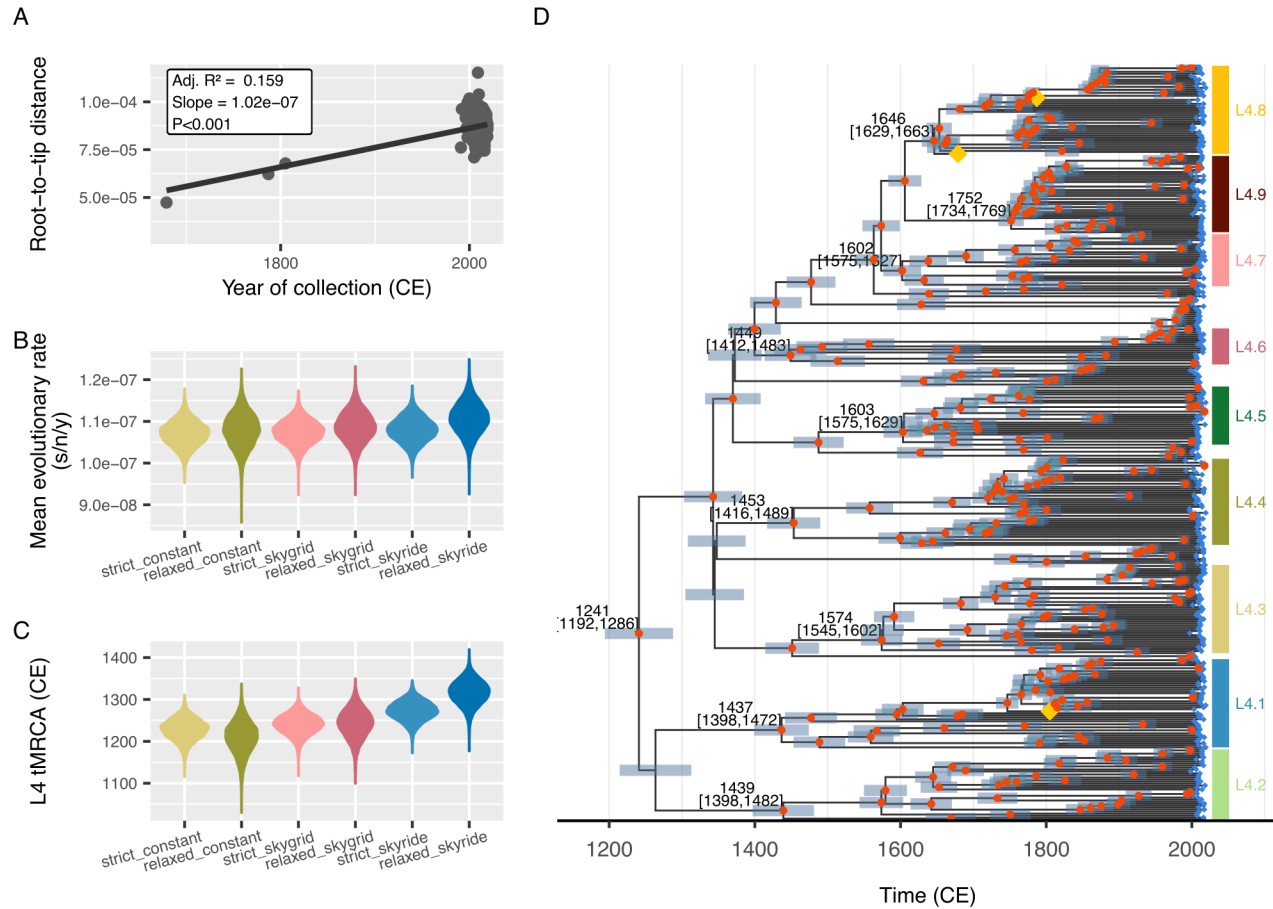

**FIG 2** Bayesian tip-dating analysis of 377 MTB L4 samples. (A) Root-to-tip regression analysis. A significant temporal signal was observed (slope = 1.02 × $10^{-7}$ s/n/y, $P$ < 0.001). (B) Violin plots of posterior distributions of mean evolutionary rate estimates computed under two clock models (strict and uncorrelated lognormal relaxed clock models) and three tree models (constant population tree model, Bayesian Skygrid coalescent tree model, and Bayesian Skyride coalescent tree model). The GTR + F + I nucleotide substitution model, identified as the best-fit model available in BEAST, was used. (C) Violin plots of posterior distributions of the tMRCA of MTB L4 as a whole, computed under the two clock models and the three tree models. (D) Time-calibrated phylogeny estimated under the best-fit Bayesian Skygrid coalescent tree model with the strict molecular clock and the GTR + F + I substitution model. Contemporary samples are marked with blue diamonds, and ancient samples are indicated by yellow diamonds. Nodes with >95% posterior clade support probabilities are indicated by orange circles. Node bars represent the 95% highest posterior density (HPD) intervals of the estimated dates. Median date estimates for major diversification events are displayed with their 95% HPD intervals.

identified the combination of the Bayesian Skygrid coalescent tree model and the strict clock model as the best-fit model (Table S3). Under this model, the mean evolutionary rate was estimated at 1.07 (1.02–1.13) × 10⁻⁷ s/n/y or 0.47 (0.45–0.50) SNPs per genome per year, comparable to the result obtained from the root-to-tip regression analysis (1.02 × 10⁻⁷ s/n/y; Fig. 2A). The basal diversification date of the entire MTB L4 was estimated to be around 1241 CE (1192–1286 CE; Fig. 2D). The fact that the strict clock model was identified as the best-fit clock model was consistent with the lack of significant evidence for the TDRP (Fig. S2) and also suggested that, likely due to the exclusion of DR-conferring mutation sites and their associated genes (see Materials and Methods), DS and DR samples in this analysis did not have significantly different evolutionary rates. Indeed, even when branches were allowed to have different rates under the uncorrelated lognormal relaxed clock model, no significant differences in evolutionary rate estimates on terminal branches leading to DS and DR samples were found (*P* value >0.05; Fig. S3). Combined, these results supported that our data preparation protocol successfully removed potential differences between DR and DS samples, at least in terms of their evolutionary rates, and that they can be analyzed together under a single strict clock. To assess the effect of data down-sampling, we repeated the analysis twice with different subsamples. All three analyses yielded comparable results (range of mean rate estimates: 1.04–1.07 × 10⁻⁷ s/n/y; range mean date estimates: 1217–1240; Fig. S4), supporting that the results were robust.

The global tree was then time calibrated under an ML framework using samples' tip dates and the rate obtained from the Bayesian tip-dating analysis above (1.07 × 10⁻⁷ s/n/y). Unlike the Bayesian tip-dating analysis above, which included only samples with relatively precise collection dates, this also included those with collection date ranges and without collection dates while accounting for collection date uncertainty (see Materials and Methods). The tMRCA of the entire MTB L4 obtained from this analysis (1180 CE, 1118–1220 CE; Fig. 3) is comparable to the one obtained from the Bayesian tip-dating analysis above (1240 CE, 1192–1286 CE; Fig. 2D). Similarly, the estimated

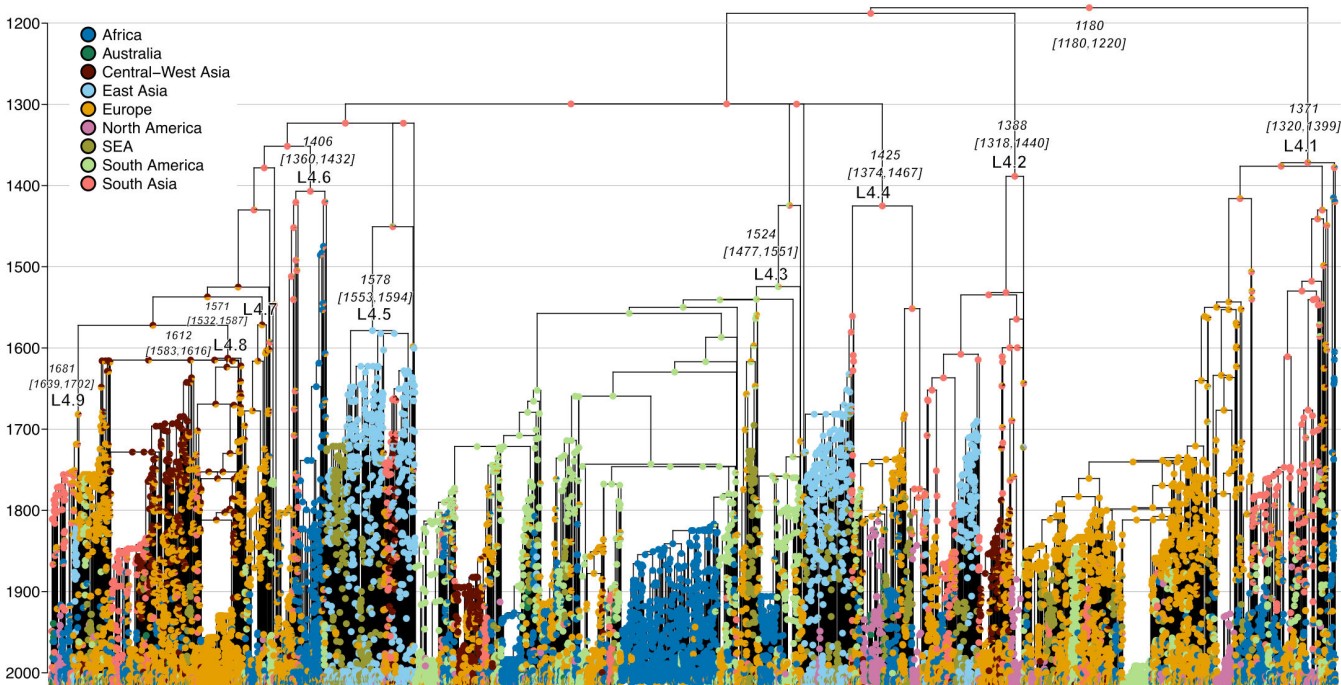

**FIG 3** Phylogeographic analysis of MTB L4. Ancestral geographical reconstruction was performed with stochastic character mapping analysis using the global time-calibrated phylogeny with the all-rates-different (ARD) model with the Fitzjohn root prior. The estimated tMRCAs, along with their corresponding 95% confidence intervals, of the nine major sub-lineages are shown. Pie charts on the tree nodes display the posterior probabilities of reconstructed geographical states.

tMRCAs of the nine major sub-lineages (Fig. 3) were also largely comparable to those obtained from the Bayesian tip-dating analysis (Fig. 2D). Overall, our results support that MTB L4 emerged and diversified between the 12th and 13th centuries.

Our rate estimates (1.02–1.13 × $10^{-7}$ s/n/y) are overall comparable to those reported by Pepperell et al. (31) (8.40–1.81 × $10^{-7}$ s/n/y) and by Roetzer et al. (32) (0.6–1.5 × $10^{-7}$ s/n/y), but they are noticeably higher than those reported by Bos et al. (8) (3.4–6.4 × $10^{-8}$ s/n/y), Kay et al. (9) (4.06–5.87 × $10^{-8}$ s/n/y), Sabin et al. (10) (0.946–1.96 × $10^{-8}$ s/n/y), and Brynildsrud et al. (12) (4.16–5.44 × $10^{-8}$ s/n/y). Interestingly, despite substantial differences in the rate estimates, our date estimates are still nevertheless similar to those reported by Brynildsrud et al. (12) (between the 10th and 13th centuries), and in contrast, other tip-dating studies, while reporting rates comparable to Brynildsrud et al. (12), tended to suggest that the MRCA of MTB L4 is much older, dating them to be between the 4th century BCE and the 11th century CE (8–10). Notably, despite using exactly the same set of historical genomes to ours to give the temporal structure to their sequence data, Sabin et al. (10) dated the MRCA of L4 to the sixth century (~1,445 years ago) in their L4-specific analysis, substantially older than our estimates. Such discrepancies could still be observed even when the clock and tree models were matched (mean tree height estimated under the constant coalescent model and the strict clock: this study, ~790.815 [743.847–841.863] years; Sabin et al. [10]: ~1567.544 [1186.119–1978.649] years; the constant coalescent model and the uncorrelated lognormal relaxed clock: this study, ~809.808 [739.274–886.478] years; Sabin et al. [10]: ~1569.051 [1054.607–2225.476] years). On the other hand, when applying our data preparation and analysis protocol to the L4-specific data set used by Sabin et al. (10), we still got a rate estimate of 1.12 (1.04–1.18) × $10^{-7}$ s/n/y and dated the MRCA of MTB L4 to 1230 (1174–1283) CE under the Bayesian Skygrid coalescent tree model and the strict clock model (see Supplemental methods), comparable to the results obtained from the analysis of our full data set. Combined, these results suggest that the observed date discrepancies were unlikely due to differences in the data sets, the tree prior models, or the clock models used. One explanation could be that they used a highly conservative data preparation method to prepare their alignment for their phylogenetic analysis, excluding any sites with missing bases (which was also adopted by Bos et al. [8]). This could potentially lead to the underestimation of nucleotide variation within the sequence data and, subsequently, the rate estimate value and, hence, overestimation of the bacterial timescale. This may explain their considerably older age estimates.

## Global transmission history of MTB L4

Stochastic character mapping analysis was performed to estimate the global transmission history of MTB L4 by using the global time-calibrated ML phylogeny and samples' tip locations. This analysis was performed using the ARD model with the Fitzjohn root prior distribution, a combination identified as the best-fit model (see Supplemental methods). The results are shown in Fig. 3 to 5.

### Early diversification and initial global spread from South Asia (12th–16th centuries)

Our analysis inferred that MTB L4 originated and diversified in South Asia, giving rise to the ancestors of several major lineages between the 12th and 14th centuries (1100–1400 CE). Following this basal diversification, MTB L4 was then inferred to spread from South Asia to other global regions during the 15th and 16th centuries (1400–1600 CE), including to Europe (L4.1) and Africa (L4.1 and L4.6). These results contrast sharply with previous hypotheses suggesting a Mediterranean origin of MTB L4 around the 4th century BCE to 4th century CE (11) or a European origin during the 10th–13th centuries (12).

South Asia has long been one of the world's most densely populated areas (33, 34), which might have facilitated early bacterial circulation and diversification. The region also historically served as a central hub for religious, cultural, diplomatic, and trading

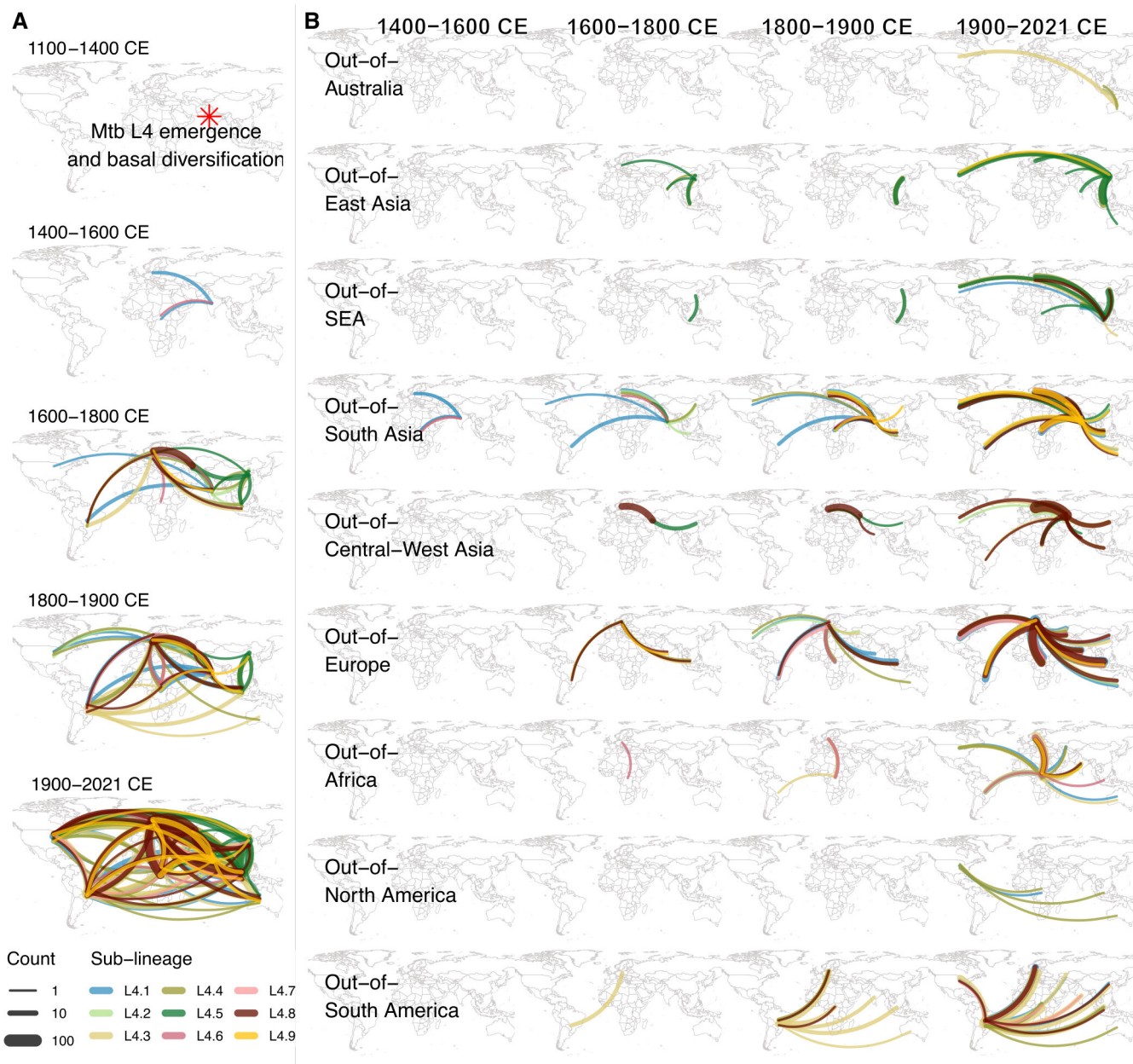

**FIG 4** Global transmission history of MTB L4 by region. Inferred transmission events of the nine L4 major sub-lineages during various time periods with all regions combined (A) and for each region separately (B). Transmission events of non-major lineages are not shown here but can be found in Fig. 3. Curved lines indicate transmission events, with the direction of transmission depicted in an anti-clockwise manner. The thickness of the curved lines represents the number of inferred transmission events, and different colors are used to distinguish between sub-lineages.

interactions, connecting together various parts of the world via both land (35–37) and maritime routes (37–40). Major trade networks existing at the time, such as the Silk Road trade route and the Indian Ocean trade network, which connected Africa, Europe, and Asia, might have contributed to these inferred early global dispersal events at least to some extent.

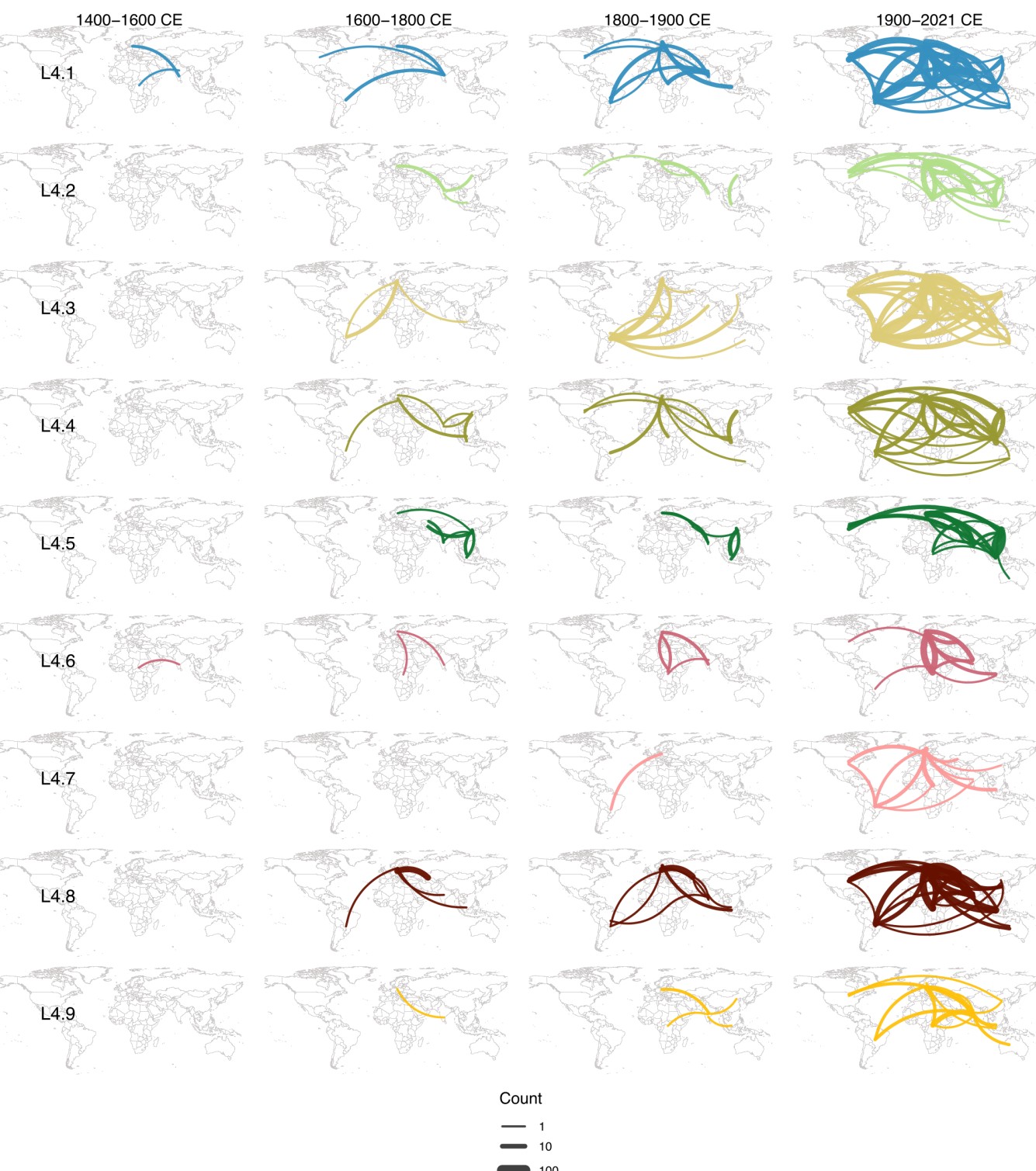

**FIG 5** Global transmission history of MTB L4 by major sub-lineage. Transmission events of non-major lineages are not shown here but can be found in Fig. 3. Curved lines indicate transmission events, with the direction of transmission depicted in an anti-clockwise manner. The thickness of the curved lines represents the number of inferred transmission events, and different colors are used to distinguish between sub-lineages.

### Extensive global expansion of MTB L4 post initial dissemination from South Asia (17th–21st centuries)

Between the 17th and 18th centuries (1600–1800 CE), South Asia remained a major source of MTB L4 transmission. However, Europe and East Asia were inferred to emerge as significant global sources as well, spreading the bacteria to all regions of the world during this timeframe following the initial introduction from South Asia in the 15th and 16th centuries. Our analysis also inferred multiple transmission events from regions outside South Asia, Europe, and East Asia during this period, including from Central-West Asia to Europe and East Asia, from Africa to Europe, and from South America to Europe. In the 19th century (1800–1900 CE), MTB L4 was inferred to continue to spread from all Old-World regions, particularly from South Asia and Europe, to both Old-World and New-World regions. A notable increase in bacterial transmission from South America to various parts of the world, particularly of the L4.3 and 4.8 lineages, was also inferred during this period.

These inferred transmission patterns broadly coincide with historical periods marked by increased global human movement, driven by expanding trade networks, industrialization, economic growth, and colonial activities centered in Europe between 1500 and 1800 CE (41, 42). The subsequent period of rapid population growth (43) and advancements in transoceanic travel (44) in the 19th century likely further facilitated the intercontinental spread of MTB L4 through growing urban populations and more frequent long-distance travel. For example, the infamous transatlantic triangular trade during the 16th and 19th centuries, one of the largest human migrations in history between Europe, Africa, and the Americas (34), might have facilitated and contributed to some of these inferred bacterial dispersal between the Old and New Worlds. These results support previous suggestions that European colonization, migration, and contact were key drivers of the global expansion of MTB (12, 45–47) while also highlighting contributions of South Asia and South America, which perhaps may have been previously under-appreciated.

During the 20th and 21st centuries, extensive global transmissions of MTB L4 across all global regions were inferred for all lineages. The age of globalization, characterized by extensive human mobility and global connectivity, likely accelerated the widespread expansion of MTB L4 during this recent time.

### Bacterial population dynamics

The overall effective population size dynamic of MTB L4 and those of various major sub-lineages were estimated by using the global time-calibrated MTB L4 phylogeny (Fig. 3) under an ML framework with a skygrid model. The effective population size of MTB L4 as a whole (Fig. 6) was inferred to steadily increase since its emergence in the early 13th century, with a rapid expansion during the 18th century, paralleling the continuous growth of the global human population as previously noted (43). However, in the mid-20th century, the bacterial effective population size was estimated to decline sharply, coinciding with the introduction of antibiotics and improvements in TB treatments (48–50). Consistent results were obtained from the initial Bayesian tip-dating analysis of the down-sampled data set of 377 sequences using the strict clock and the Skygrid coalescent tree model (Fig. S5).

Interestingly, at the major sub-lineage level, the effective population sizes of all sub-lineages were independently inferred to show strikingly similar dynamics, paralleling the overall pattern (Fig. 6; Fig. S6)—all were inferred to continuously increase from their emergences (ranging between 14th and 17th centuries) until the mid-20th century, when a decline occurred. These results suggest that all major sub-lineages experienced broadly similar patterns of selection pressure through time, particularly the negative selection pressure imposed by antibiotics and improvements in TB treatments around the mid-20th century. Different peak effective population sizes could be due to ascertainment bias, varying intrinsic transmissibility, and/or differences in geographical, biological, and host-related factors. The interplay between these factors and their

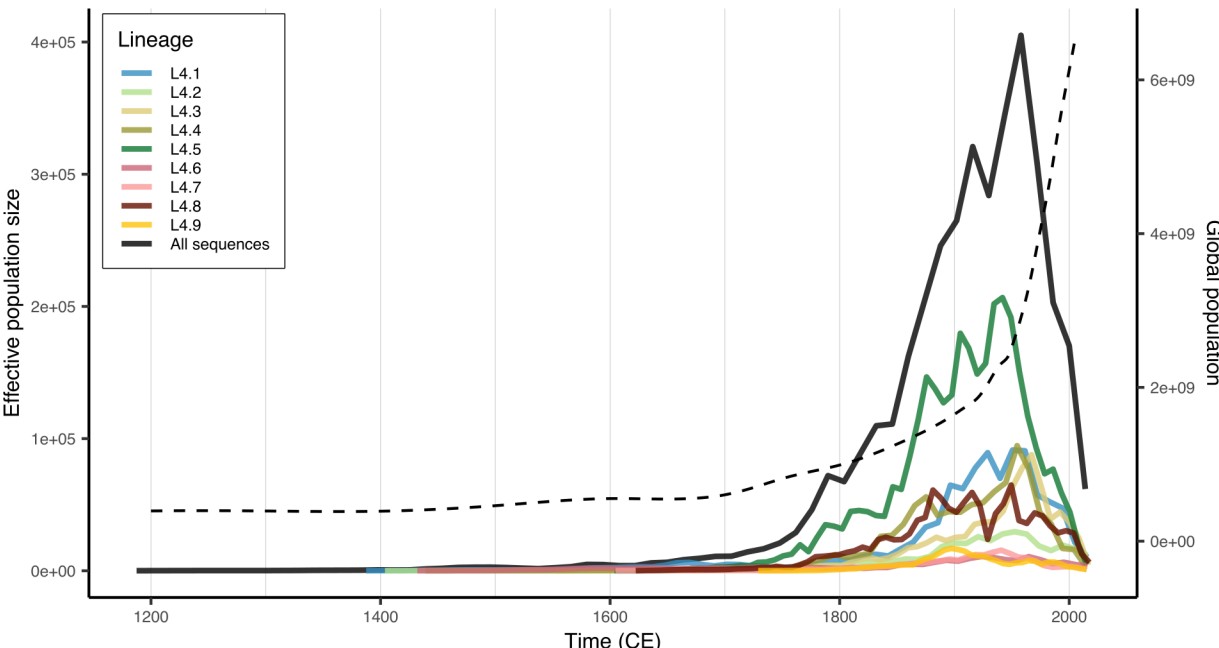

**FIG 6** MTB L4 effective population size dynamics. Population dynamics were estimated for the entire data set (black line) and for the nine major sub-lineages (colored lines). The dashed line represents the estimated global human population size during 1200 CE–2015 CE, reported by Roser et al. (43).

contribution to shaping the population dynamics of individual major sub-lineages remains unclear and warrants further investigation.

## Conclusions

Understanding the past transmission history of MTB L4 is crucial for understanding its current global distribution. Our phylogenetic analysis, which, to our knowledge, is the most comprehensive one to date, suggests that MTB L4 likely originated and diversified in South Asia around 700–900 years ago (12th and 13th centuries), giving rise to various ancestral sub-lineages before spreading to other parts of the world. These findings suggest a potentially pivotal role of South Asia, with its historical trade networks and maritime connections, in promoting the early intercontinental spread of MTB L4.

Our results challenge some existing hypotheses regarding the origins of MTB L4, including the Mediterranean (i.e., African and European) origin around the fourth century BCE to fourth century CE proposed by O'Neill et al. (11) and the European origin during the 10th–13th centuries proposed by Brynildsrud et al. (12). These earlier hypotheses were based on analyses of relatively limited data sets (O'Neill et al. [11]: Old-World MTBC data collection, 552 samples from 51 countries, 143 of which were L4 from 20 countries; Brynildsrud et al. [12]: 1,207 samples from 15 countries), and phylogeographic inference is inherently highly sensitive to data availability. Phylogeographic inference estimates probable geographical locations at all internal nodes/branches within a phylogeny based on the sampling locations of samples at the tips and the estimated branch lengths. Therefore, the result can be biased if the samples analyzed are largely incomplete or unrepresentative of the true global geographical diversity, simply due to the lack of signals from the tips. Additionally, differences in timescale calibration methods can also influence estimated divergence times, which in turn can affect the accuracy of estimation of transition rates between locations. Compared to previous studies, our analysis used a significantly larger and more geographically comprehensive data set of MTB L4, including 11,154 samples from 106 countries (Fig. 1). Combined with a systematic approach to evolutionary timescale estimation, we believe that our analysis provides a data-driven and justified updated MTB L4's evolutionary history and geographic origins.

Besides South Asia, as previously suggested (12, 45, 47, 51), our findings support that European exploration and colonization likely played a big part in the subsequent global dissemination of various MTB L4 sub-lineages. We also observed temporal links between the increase in global connectivity and the increase in intercontinental transmission of the bacteria during the 17th–21st century. All of these highlight the profound impact of anthropological changes on the trajectory of bacterial transmission.

Our population dynamics analysis suggests a steady increase in the overall MTB L4 population size from their basal diversification event onward until the age of antibiotics and modern medicine in the mid-20th century, when the inferred bacterial population size dropped sharply. Remarkably, similar patterns were independently observed across all major sub-lineages, highlighting the significant contribution and success of antimicrobial treatment and modern medicine in TB control at large.

By integrating phylogenetic, demographic, and phylogeographic analyses, this study contributes to a more refined understanding of MTB L4's evolutionary history. However, we should emphasize that results presented here represent the most likely estimates based on currently available data and methodological methods. As more genomic and epidemiological data become available and analytical techniques continue to evolve, future research may provide further insights, leading to a more complete and refined understanding of MTB L4's evolutionary history and its close relationship with their human hosts.

## MATERIALS AND METHODS

### Compilation of MTB L4 WGS data

We compiled 12,496 accession numbers of whole-genome short-read sequences of contemporary MTB L4 samples reported worldwide from research publications and public repositories (NCBI SRA and EBI-EMBL ENA). Data compilation was performed up until August 2024. Raw sequence reads were downloaded, excluding samples without known geographical origins ($n = 407$). The data set was curated to deduplicate and exclude samples with overall read-mapping coverage below 90% and depths below 10×, resulting in a clean data set of 11,179 high-quality contemporary samples (see Supplemental methods for more details). mtbtyper (https://github.com/ythaworn/mtbtyper) was used for taxonomic identification, and TB-Profiler (52) was used for DR genotyping, both with default settings.

### Joint variant calling

Joint variant calling was performed by using snpplet (https://github.com/CENMIG/snpplet) to generate a multiple sequence alignment from Genome Variant Call Format (GVCF) files, including the H37Rv reference genome, and one L1 sample (SRR8375623) serving as an outgroup. Filters were applied to remove sites with a low probability of actually being a variant site ("QualByDepth" or "QD" score < 2), sites with a low mapping score ("Mapping Quality" or "MQ" score < 40), indel sites, sites within the regions of "refined low confidence and low pileup mappability" as described in reference (53), which included such as sites in repetitive PE/PPE genes, sites within drug resistance genes, and known drug resistance mutation sites (54). The final clean multiple sequence alignment contains 356,934 sites across 11,184 samples, comprising 11,179 contemporary samples, 3 ancient samples, 1 H37Rv reference sample, and 1 L1 sample.

### Global phylogenetic reconstruction

An ML global phylogeny was estimated using IQ-TREE2 (19) with the GTR + I + Γ(4) model, corrected for the nucleotide frequencies of constant sites. Branch support was computed using 1,000 bootstrap trees estimated using the ultrafast bootstrap approximation method (55). The tree was rooted with the L1 sample and was subsequently removed. Samples with unusually long terminal branches ($n = 25$) were detected by

using TreeShrink (56) with a false-positive tolerance rate of 0.10 and were removed. Samples with unexpectedly short terminal branches, given their recent collection dates ($n = 3$), were also dropped from the tree. The H37Rv reference genome clustered with other L4.9 samples as expected but was also removed from the tree due to its uncertain collection date and location. The final clean data set forming the basis of all of our subsequent analyses thus consisted of 11,154 samples of MTB L4, including 11,151 contemporary samples and 3 ancient samples (Table S1).

## Evolutionary rate and timescale estimation

The rate and timescale of MTB L4 evolution were estimated in two steps. First, we conducted a tip-dating analysis with BEAST v1.10.4 (57). To make the computation feasible, this initial Bayesian tip-dating analysis was conducted on a down-sampled data set of 377 sequences (Table S2; see Supplemental methods for more details on the down-sampling). A multiple sequence alignment of the down-sampled sequences was generated, and their ML phylogeny was then estimated using the same procedure as described in "Global phylogenetic reconstruction." Tip dates of ancient samples were assumed to be their human host's year of death, while sampling years were used for contemporary samples. The root-to-tip regression analysis suggested that there was a sufficiently strong temporal signal in the data set to allow for Bayesian tip dating and rate estimation (slope = $1.02 \times 10^{-7}$ s/n/y, $P$ value < 0.001). In addition, the data set was assessed for the TDRP by computing evolutionary rate estimates over various timescales by using LSD2 (58) and examining if there was a significantly negative correlation between the rate estimates and the measurement timescales (29, 30).

Three tree prior models were examined in the initial Bayesian tip-dating analysis: (i) the constant population size tree model, (ii) the Bayesian Skyride coalescent tree model, and (iii) the Bayesian Skygrid tree model, in combination with two molecular clock models: (i) the strict clock model and (ii) the uncorrelated lognormal relaxed clock model. All analyses were done using the GTR + F + I nucleotide substitution model, identified as the best-fit model among all models supported by BEAST (57) as determined under the Bayesian information criterion with ModelFinder (59) implemented in IQ-TREE2 (19). For all analyses, the Markov chain Monte Carlo sampling length was set to $1.0 \times 10^9$ steps, and the parameter values were logged every 100,000th step. BEAST XML files were manually edited to account for the number of bases in constant sites. Tracer (60) was used to inspect the results to ensure trace mixing and convergence. The best-fit combination of the tree and clock model was determined based on marginal likelihood scores, computed using the path-sampling and stepping-stone methods implemented in BEAST v1.10.4 (57). To assess the robustness of our evolutionary rate and date estimates, the analysis was repeated with two independently down-sampled data sets under the best-fit prior settings (see Table S2 for the lists of sequences used and Supplemental methods for more details).

We subsequently used the obtained rate estimate ($1.07 \times 10^{-7}$ s/n/y) to inform the tip-dating analysis of the global MTB L4 tree ($n = 11,154$) under an ML framework with LSD2 (58), which was much less computationally expensive. This tip-dating analysis also included samples with collection date ranges and those without collection dates, accounting for their collection date uncertainty. Lastly, to further validate our results, we applied our data preparation and Bayesian tip-dating analysis protocol to the L4-specific data set analyzed by Sabin et al. (10) to estimate the rate and timescale of MTB L4 (see Supplemental methods).

## Phylogeographic analysis

The geographical origin and global transmission history of MTB L4 was inferred through stochastic character mapping analysis using the global time-calibrated ML tree and the geographical regions of the samples. The analysis was performed with the phytools package (61) in R (62). The ARD model with the Fitzjohn root prior was identified as the best-fit model based on the Akaike Information Criterion (Table S4) and was therefore

used in the final analysis (see Supplemental methods for more details). The final analysis was conducted using the simmap function in phytools (61) with 1,000 simulations, each employing a state transition matrix sampled from the posterior distribution of the model (Q = "mcmc"). Using a Python script described in a previous study (63), transmission events across regions over time were counted from the obtained phylogeny, where the state of each internal node was chosen to be the region with the highest posterior probability.

## Reconstruction of the bacteria's past population dynamics

The past population dynamics of the entire MTB L4 and of each major sub-lineage were estimated using the mlesky package (64) in R (62) with the mlskygrid function. The smoothing parameter (tau) was set to 60. The number of grid points (res) was set to 60, determined as the optimal value using the optim_res_bic function in the mlesky R package (64), given the tau value. For the nine major sub-lineages, pruned time-calibrated phylogenies were used as input.

## ACKNOWLEDGMENTS

We thank the Center of Excellence in AI-Based Medical Diagnosis (AI-MD), Mahidol University, for providing computational resources. We are also grateful for helpful comments from Francesc Coll.

This research project is supported by Mahidol University (MU's Strategic Research Fund): fiscal year 2023 (grant number MU-SRF-WC-02B/66).

P.A. conceptualized, designed, and supervised the project. B.J. gathered and analyzed data. B.J. and P.A. wrote the original draft. P.P. and P.A. acquired funding and administered the project. All reviewed and discussed the results.

## AUTHOR AFFILIATIONS

[1]Department of Microbiology, Faculty of Science, Mahidol University, Ratchathewi District, Bangkok, Thailand
[2]Pornchai Matangkasombut Center for Microbial Genomics, Department of Microbiology, Faculty of Science, Mahidol University, Bangkok, Thailand

## AUTHOR ORCIDs

Bharkbhoom Jaemsai  https://orcid.org/0009-0002-4908-6469
Pakorn Aiewsakun  http://orcid.org/0000-0002-5665-4041

## FUNDING

| Funder | Grant(s) | Author(s) |
|---|---|---|
| Mahidol University (MU) | MU-SRF-WC-02B/66 | Prasit Palittapongarnpim |
| | | Pakorn Aiewsakun |

## DATA AVAILABILITY

All relevant data are within the paper and its supplemental maerial.

## ADDITIONAL FILES

The following material is available online.

### Supplemental Material

**Supplemental material (mSystems00427-25-s0001.docx).** Supplemental methods and Figures S1 to S6.

**Supplemental tables (mSystems00427-25-s0002.xlsx).** Tables S1 to S4.

Open Peer Review

**PEER REVIEW HISTORY (review-history.pdf).** An accounting of the reviewer comments and feedback.

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
