## [Reviewer comments · mSystems]

South Asian origin and global transmission history of *Mycobacterium tuberculosis* Lineage 4

Bharkbhoom Jaemsai, Prasit Palittapongarnpim, and Pakorn Aiewsakun

Corresponding Author(s): Pakorn Aiewsakun, Mahidol University Faculty of Science

Review Timeline:

Submission Date:

April 1, 2025

Accepted:

April 17, 2025

Editor: David Cleary

Reviewer(s): The reviewers have opted to remain anonymous.

Transaction Report:

DOI: <https://doi.org/10.1128/mSystems.00427-25>

Re: mSystems00427-25 (South Asian origin and global transmission history of *Mycobacterium tuberculosis* Lineage 4)

Dear Dr. Pakorn Aiewsakun:

Your manuscript has been accepted, and I am forwarding it to the ASM production staff for publication. Your paper will first be checked to make sure all elements meet the technical requirements. ASM staff will contact you if anything needs to be revised before copyediting and production can begin. Otherwise, you will be notified when your proofs are ready to be viewed.

Sincerely,
David Cleary
Editor
mSystems

Reviewer #1 (Comments for the Author):

I commend the authors on the additional work for this paper. Although the dating remains contentious, as Mtb dating always will be, I feel the authors have qualified their analyses sufficiently for publication.